# Complex Sleeve Lobectomy Has Lower Postoperative Major Complications Than Pneumonectomy in Patients with Centrally Located Non-Small-Cell Lung Cancer

**DOI:** 10.3390/cancers16020261

**Published:** 2024-01-06

**Authors:** Luca Voltolini, Domenico Viggiano, Alessandro Gonfiotti, Sara Borgianni, Giovanni Mugnaini, Alberto Salvicchi, Stefano Bongiolatti

**Affiliations:** 1Thoracic Surgery Unit, Careggi University Hospital, 50134 Florence, Italy; luca.voltolini@unifi.it (L.V.); g.mugnaini12@gmail.com (G.M.);; 2Department of Experimental and Clinical Medicine, University of Florence, 50134 Florence, Italy

**Keywords:** locally advanced non-small-cell lung cancer (NSCLC), bronchoplasty, complex sleeve lobectomy, pneumonectomy, survival

## Abstract

**Simple Summary:**

For centrally located non-small cell lung cancer (NSCLC), sleeve lobectomy would be preferable to pneumonectomy (PN). In this context, however, the role of complex sleeve lobectomy (CLS) is still poorly understood. Our study compared CLS with PN using a retrospective analysis. Research shows that CLS has a lower 90-day mortality rate and fewer major complications compared to PN, while oncological outcomes remain comparable. These results support the contention that CLS is a safe and effective procedure for centrally located NSCLC, even after neoadjuvant treatment.

**Abstract:**

Background: Standard sleeve lobectomies are recommended over pneumonectomy (PN), but the efficacy and oncological proficiency of complex sleeve lobectomies (CSLs) have not been completely investigated. The aim of this study was to report our experience in CSL in patients affected by a centrally located non-small-cell lung cancer (NSCLC), comparing all the variables and outcomes with PN. Methods: From 2014 to 2022, we collected the data of patients who underwent PN and CSL for NSCLC, excluding neuroendocrine tumors, salvage surgery or carinal resection. Regression analysis was used to assess the association between procedures and complications; the Kaplan–Meier method and Cox regression analysis were used to evaluate survival and risk factors of reduced survival. Results: We analyzed *n* = 38 extended sleeve lobectomies and *n*= 6 double-sleeve lobectomies (CSL group) and *n*= 60 PNs. We had a trend toward higher postoperative mortality in the PN group (5% vs. 0%, *p* = 0.13). Major complications and bronchial fistula developed in 21.7% and 6.8% (*p* = 0.038) and in 6.7% and 4.5% (*p* = 0.64), respectively. The right side was identified as risk factor for major complications, whereas age > 70 and PN had a trend of association in multivariable analysis. The median OS was similar between the two groups (*p* = 0.76) and cancer recurrence was the only significant risk factors of reduced OS. Excluding functionally compromised patients, the OS of CSL was better than that of PN (67% vs. 42%, *p* = 0.25). Conclusions: Considering that major complications are often associated with mortality after surgery for centrally located NSCLC, CSLs could be considered an alternative to PN while also ensuring comparable survival.

## 1. Introduction

Surgical treatment of locally advanced non-small-cell lung cancer (NSCLC) with hilar involvement often requires pneumonectomy (PN), which is an effective oncological treatment but comes at the price of high morbidity and mortality, which also leads to significant impairment of lung function and quality of life and limits the possibility of adjuvant treatments, which are often necessary [1,2]. The classic standard sleeve lobectomy (SSL) is nowadays recommended over PN if it is anatomically suitable and a margin-negative resection can be achieved [3], as several studies have shown favorable outcomes after sleeve resection in terms of morbidity and mortality, but also overall survival, recurrence rate and disease-free survival [4,5,6,7,8,9]. Based on these results, the SSL/PN ratio has been proposed as a valid index to determine the quality standard and performance in a specialized thoracic department [10]. In selected centrally located NSCLCs, SSL is not sufficient to achieve complete resection, and then an extended sleeve lobectomy (ESL, defined as atypical bronchoplasty with resection of more than one lobe) or a double-sleeve lobectomy (sleeve bronchial resection associated with sleeve arterial resection) would be required, to avoid PN, but these procedures are rare and technically more challenging [11] due to the different bronchial caliber, fragility of the distal bronchial and vascular stumps, and potentially increased tension at the anastomotic site [12]. These intraoperative problems could affect the postoperative course with a theoretically increased risk of bronchial and vascular anastomoses such as bronchial fistulas and/or stenoses as well as arterial and venous thrombosis [13,14,15,16,17]. Although SSL is now an accepted surgical procedure, the safety, efficacy and oncologic performance of complex sleeve lobectomy (CSL) have not been fully investigated, especially in the context of neoadjuvant therapy, which could contribute to jeopardizing the airway healing already compromised by the aforementioned features. Moreover, very few studies have directly compared the efficacy of these procedures with PN [13,14,16,17].

The aim of this study was to report our institutional experience with CSL, defined as ESL or double-sleeve lobectomy, in patients with centrally located NSCLC, and to compare the short-term (overall complications, major complications, mortality) and long-term (overall and disease-free survival) outcomes with those of a contemporary cohort of PN patients, focusing on the risk factors for postoperative complications, including the influence of neoadjuvant treatment.

## 2. Materials and Methods

### 2.1. Patient Population

Among 2368 consecutive patients with primary NSCLC who underwent major lung resection from January 2014 to January 2022, we selected and analyzed data from bronchoplastic resections and PNs. Bronchoplastic resections are considered CSLs, including ESLs (atypical bronchoplasty with resection of more than one lobe) and standard double-sleeve lobectomies (broncho-vascular sleeve lobectomy), according to Inci et al. [18]. Patients with neuroendocrine tumors or diseases other than NSCLC, patients treated with carinal sleeve resection and patients who underwent surgery after curative-intent chemoradiotherapy (salvage surgery) or a simpler pulmonary angioplasty technique such as tangential suture of the pulmonary artery were excluded from this analysis.

The primary end points were the following:To evaluate and analyze the incidence of the overall complication rate, major complication and mortality by comparing the CSL group with the contemporary PN group;To evaluate and analyze the risk factors of major complications as defined as 3b or more in the Clavien–Dindo classification [19].

The secondary end points included the following:The evaluation of overall and disease-free survival between the two groups;Analysis of the risk factors of poor overall survival.

Our institutional review board granted approval and waived the requirement for specific informed consent for this retrospective study.

The preoperative examination included contrast-enhanced whole-body computed tomography (CT) and 18F-fluorodeoxyglucose whole-body positron emission tomography (PET-CT). Preoperative bronchoscopy was performed in all patients to assess bronchial tree involvement and for diagnostic purposes. In the case of suspicious mediastinal lymph nodes on CT or PET-CT scans, patients underwent cytological or histologic examination with endobronchial or esophageal ultrasound or video-mediastinoscopy. After a multidisciplinary tumor board discussion, patients with histologically proven N2 disease received neoadjuvant chemotherapy, unless only one station was affected. In this case, the patient could receive upfront surgery followed by adjuvant chemotherapy [20]. Preoperative assessment included arterial blood gas analysis and spirometry, pletysmography and measurement of diffusion capacity for carbon monoxide, and in patients with impaired lung function a ventilation/perfusion scan and/or exercise test was performed. Patients were defined as “impaired” if relevant comorbidities were present (chronic heart disease with impaired function, chronic renal failure requiring dialysis, liver failure with coagulopathy, stroke or other vascular disease with deep impact on mobility) and/or FEV1% < 60% and/or DLCO% < 50%. Postoperative complications and mortality, defined as any death within 90 days after surgery or during the same hospital stay, were analyzed as well as overall survival (OS) and disease-free survival (DFS). Local recurrence was defined as recurrence in the preserved lobe or bronchovascular structures; regional recurrence was defined as recurrence in homolateral lobe/s other than the preserved lobe, in the pleural space, hilar or mediastinal lymph nodes; distant recurrence was defined as any metastasis developing in extra-thoracic organs or in the contralateral lung. All patients completed follow-up and were included in the survival analysis. The last follow-up examination took place in December 2022.

### 2.2. Operative Technique

Our strategy (inclination) was to perform CSL whenever technically feasible, even in patients with adequate pulmonary reserve who could potentially tolerate PN [13]. The surgical approach was through a muscle-sparing posterolateral thoracotomy and bronchial anastomosis was performed with a 4.0 polypropylene (PLP) single running suture without routine wrapping [13]. The vascular anastomosis was performed with a 5.0 PLP double-arm running suture [13]. After CSL, two large-bore chest tubes were inserted to help the remaining parenchyma to fill the pleural cavity. Routine bronchoscopy was performed at the end of surgery, before hospital discharge and whenever we suspected atelectasis, broncho-pleural fistula (BPF) or persistent air leak.

### 2.3. Statistical Analysis

Statistical analysis was performed using SPSS 24.0 software (IBM SPSS Statistics for Machintosh, Version 24.0. IBM Corp.: Armonk, NY, USA). Continuous variables, if normally distributed, were expressed as means and standard deviations and compared with unpaired Student’s *t*-test results. Categorical variables were calculated as percentages and were analyzed using the χ^2^ test or the Fisher exact test as appropriate.

To identify preoperative risk factors of postoperative mortality and morbidity, univariate and multivariable logistic regression analyses were performed on selected clinical variables (age, sex, type of procedure, Eastern Cooperative Oncology Group (ECOG) performance status, FEV1 < 60%, DLCO < 50%, Charlson Comorbidity Index, neoadjuvant chemotherapy, clinical stage, operation side). Variables with a *p* value less than 0.2 at the univariate analysis were entered into the multivariable model.

The Kaplan–Meier method was used to estimate OS and DFS. Overall survival was calculated from the date of operation to death or the date of the last follow-up (December 2022); DFS was calculated from the date of surgery to the date of the first cyto-histologically proven evidence of recurrence or death. Follow-up was assessed at outpatient visits including interval medical history, physical examination and enhanced contrast whole-body CT scan every six months. Differences in OS and DFS between groups were evaluated using log-rank analysis. Univariate and multivariable Cox regression analyses were used to assess risk factors of reduced OS and DFS. The significance level was defined as *p* < 0.05.

## 3. Results

### 3.1. Preoperative Characteristics

During the study period, 133 (5.6%) bronchoplastic resections and 60 (2.5%) PNs were performed. Among bronchoplastic resections, 44 (33%) were considered CSLs including 38 ESLs and 6 standard double-sleeve lobectomies.

Preoperative patient characteristics are showed in Table 1.

The two groups were similar in terms of demographic and functional parameters. Eleven (25%) of the patients in the CSL group had impaired cardiopulmonary function that precluded or strongly discouraged PN due to potentially excessive perioperative risk.

Squamous cell carcinoma was the most common histologic pattern in both groups (51.7% vs. 65.9%) and the distribution of clinical tumor, node and distant metastasis (TNM) stage was not statistically different between the two groups, with most patients having clinical stage III NSCLC. Patients with locally advanced NSCLC were treated preoperatively with chemotherapy (35% vs. 31.8%) or chemoradiotherapy (0%, 4.9%) in the PN and CSL groups, respectively.

### 3.2. Type of Resections

According to the modified Okada classification [12,14], 15 cases of type A ESL (resection of right upper plus middle lobe ± segment 6), 2 cases of type B (resection of left upper lobe + segment 6), 17 cases of type C (resection of left lower lobe + lingulectomy), 3 cases of type D (lower bilobectomy and anastomosis between the right upper and main bronchus) and 1 not classified (right upper lobectomy and superior segmentectomy of the lower lobe with anastomosis between the right main bronchus and basal segments plus middle lobe bronchus) CSL were performed as summarized in Table 2.

Concomitant pulmonary angioplasty was performed in 10 (26.3%) patients, including pulmonary artery sleeve resection in 9 patients and pulmonary venoplasty in 1. Pulmonary artery sleeve resection was common in type A (7/15, 46.6%) and B (2/2, 100%) and never necessary in type C ESL. Six patients underwent standard double-sleeve resection: three left upper, one left lower and two upper right sleeve lobectomies. In the PN group, the majority of patients underwent left PN (61.7%). The covering of the bronchial stump was predominantly used in patients undergoing PN (55% vs. 11.4%, p < 0.01). Differently from our previous experience [10], in the last 16 cases there was no conversion from CSL to PN during surgery.

### 3.3. Postoperative Results

The postoperative results in the two groups are depicted and compared in Table 3.

Although not significant, we observed a trend towards higher postoperative mortality in the PN group (5% vs. 0%, *p* = 0.13): three deaths were recorded in the PN group within 90 days of surgery, with all three patients dying after right PN complicated by the onset of acute respiratory distress syndrome (ARDS) in two and fatal broncho-pleural fistula, empyema and sepsis in the last.

Major postoperative complications, categorized as 3b or more according to the Clavien–Dindo classification [19], developed in 21.7% and 6.8% of patients, respectively (*p* = 0.038). Bronchial fistula and empyema were observed in 6.7% and 4.5%, respectively (*p* = 0.64). In the CSL group, we had two anastomotic leaks, which were treated conservatively with thoracostomy in one case and CT-guided small-bore chest drainage near the bronchial fistula in the other case; no patient experienced complications related to vascular reconstruction. The percentages of patients experiencing at least one postoperative complication were 60% and 54.5% in the two groups (*p* = 0.57), and the mean hospital stay was 10.68 ± 7.1, 12.7 ± 8.93 (*p* = 0.18). The univariate analysis demonstrated PN and right side as significant risk factors for major postoperative complications, but only the right side (OR 5.75, CI95% 1.43–23.1, *p* = 0.014) was confirmed as significant in the multivariable model (Table 4).

Neoadjuvant therapy (OR 2.06, *p* = 0.39), complex sleeve resection (OR 1.5, *p* = 0.64) and bronchial flap coverage (OR 0.33, *p* = 0.32) showed no significant association with bronchial complications.

Stage distribution, pathological N-status, tumor diameter and resection status were comparable between the two groups. Complete resection was achieved in 88.3% and 88.6%, respectively, with R1 status defined by involvement of the uppermost mediastinal lymph node station in all cases; no patient had residual tumoral tissue at the bronchial, vascular or parenchymal margins (Table 3).

### 3.4. Oncological Outcomes

At a median follow-up of 24 (4–89) months in the PN group and 18 (6–75) months in the CSL group, there was no statistically significant difference in the 3-year (54% vs. 47%) and 5-year overall survival rates (42% vs. 42%, *p* = 0.76). The median OS also did not differ between the two groups (Figure 1): 45 months (95%CI 14.4–74.5) for PN and 35 (95%CI 6.85–63.12) for CSL. Median DFS was also similar between the two groups (Figure 2), specifically 24 months for PN (95%CI 4.74–43.2) and 22 for the CSL group (95%CI 12.1–31.8); the 3-year DFS rates were 41% and 38%, respectively (*p* = 0.72).

Consequently, no difference in non-cancer mortality was observed between the two groups, with overall recurrence rates of 43.3% and 45.5% (*p* = 0.83) for PN and CSL, respectively. However, patterns of recurrence differed between the two groups with a higher rate of local recurrence in the CSL group (13.6% vs. 1.7%), while cancer recurrence was predominantly distant in the PN group (36.7% vs. 27.3%) (Table 3).

A further analysis of OS with the exclusion of compromised patients submitted to CSL demonstrated a non-significant survival advantage with a better 3-year OS (67% vs. 54%, *p* = 0.25) in comparison with PN patients (Figure 3).

The univariate analysis of OS showed that poor performance status, cancer recurrence and adjuvant radiotherapy were significantly associated with reduced OS, but at the multivariable Cox logistic regression analysis (Table 5) cancer recurrence (HR 5.5 CI95% 2.14–14.1, *p* < 0.01) was the only significant risk factor of decreased OS.

## 4. Discussion

Pneumonectomy has been the gold standard for centrally located NSCLC for years, but this procedure is associated with significant postoperative morbidity and mortality, severe lung function impairment, lower compliance with adjuvant treatments and reduced quality of life (QoL) [21,22]. With increasing experience in parenchymal sparing resection, classical standard sleeve lobectomy, when anatomically feasible, has nowadays replaced PN in locally advanced NSCLC due to its better postoperative outcomes with comparable oncological outcomes [6,7,8]. A further step towards the parenchymal-sparing technique is CSL, defined in our study as ESL or double-sleeve lobectomy. However, there is limited evidence in the literature to support the role of these procedures in locally advanced NSCLC, particularly with regard to their technical challenges with the potential risk of catastrophic postoperative complications, including pulmonary vascular thrombosis and disruption of bronchial anastomosis [12,13,14,15,16,17]. In this study, CSL appears to be a safe and effective surgical procedure for centrally located NSCLC with superior postoperative results (in terms of lower mortality and a lower major complication rate) and equivalent long-term outcomes (OS and DFS) compared to PN (Table 6). We have chosen to group ESL and double-sleeve lobectomy together because these procedures are technically more demanding than SSL or PN and are performed for centrally located tumors with aggressive and infiltrative behavior that require extensive resection, making them a true alternative and comparison to PN. To avoid historical bias, we included patients who underwent PN and CSL in the same period (2014–2021) to ensure a homogeneous population treated by the same surgical group with the same indications, experience with surgical techniques and patterns of care. In our analysis, we observed a lower 90-day mortality for CSL, although this was not statistically significant compared to PN. Despite the complexity of these cases, in which 32% (n = 14) of patients had clinical N2 disease and 36.3% (n = 16) underwent surgery after neoadjuvant therapy, we did not observe 90-day mortality in the CSL group. In contrast to our previous experience [10], no patient scheduled for CLS had to be switched to PN in the second period (2020–2022). The negative impact of PN on the patients’ postoperative course is also evident from our analysis of postoperative complications. Although the incidence of postoperative complications did not differ between the two groups, according to the Clavien–Dindo classification, they were more severe in patients treated with PN and include ARDS, bleeding requiring surgery and bronchopleural fistulas, which are life-threatening complications, especially in patients with only one lung.

In the group of CSL patients, the majority of patients who experienced complications suffered from prolonged air leak, bronchial obstruction requiring bronchoscopy, atrial fibrillation and anemia with blood transfusion; these adverse events are not life-threatening and can be easily managed, although they may result in prolonged hospital stays. Regarding the technical complexity of CSL and the resulting potentially fatal complications associated with bronchial and/or vascular reconstruction, there are controversial findings in the literature [23]. Okada et al. [12] were the first who reported the postoperative clinical and oncological results of n = 15 ESL patients at the end of the 1990s, showing no postoperative death, acceptable complications, no incidence of broncho-pleural fistula or anastomotic stenosis and complete resection achieved in all patients. Others [14,15,16,23] reported a low mortality rate (0–3%), acceptable overall and anastomotic complications (3–8.7%), a high rate of complete resection and long-term oncological outcomes comparable to those of the control group, which consisted of patients undergoing PN [14] or SSL [16]. Conversely, two other recent papers [16,17] reported four and two cases of completion of PN due to venous thrombosis (n = 2), arterial thrombosis (n = 1) or bronchopleural fistula (n = 3), respectively. In our series, we did not observe any bronchial or vascular complications that required a final PN: the two cases of BPF healed quickly with conservative treatment. Vascular complications are more frequently reported after type A and B ESL [14,15,16] and could be prevented with some surgical maneuvers such as complete mobilization of the hilum, U-shaped pericardial release and sometimes the transposition of the inferior pulmonary vein to the superior pulmonary vein [17]. Although the different caliber and greater fragility of the distal bronchial stumps place the bronchial anastomosis at risk of rupture, we do not routinely cover the bronchial anastomosis and reserve this maneuver for patients with established conditions that have altered local blood supply, such as diabetes mellitus or preoperative chemoradiation treatments [13,24,25,26,27,28]. Our series of CSLs showed an acceptable incidence of bronchopleural fistulas (4.5%), regardless of whether the anastomosis was covered or not.

Regarding the long-term outcomes of CSLs, we have demonstrated the oncological adequacy of these procedures and achieved similar 3-year DFS and OS rates between CSL and PN. Recently, Hattori et al. [17] showed that OS and DFS differed between ESL and PN depending on the side: in particular, right PN had significantly worse outcomes than right ESL, while there were no differences in long-term survival between ESL and PN on the left side, suggesting that the parenchymal-sparing procedure is particularly effective for the right side and not as important for left tumors. We did not find such a striking difference in relation to the surgical side, even though all three deaths occurred in patients who underwent PN on the right side.

The main concern with CSL is the potentially higher rate of local recurrence compared to PN. The local recurrence rate after ESL is reported in recent publications to be rare, ranging from 0 to 8% [8,9,10,11,12,13,14,15]. Compared to these reports and to PN, we observed a higher incidence of local recurrence in the CSL group (13.6% vs. 1.6%). However, we consider this result acceptable considering that in our series two-thirds of the patients were stage III (large tumor with hilar or mediastinal lymph node metastasis). Furthermore, thanks to lung-sparing surgery, a large proportion of patients (47.7%) who underwent CSL were able to receive adjuvant therapy, even though a consistent proportion of patients (n = 11, 25%) could not tolerate PN. This resulted in considerable equivalence in median (24 vs. 21 months) and 3-year DFS rates (40.5% vs. 30%) between PN and CSL in relation to the different pattern of recurrence in the two groups.

The main limitations of this study are the potential and inherent biases of a single institutional retrospective study, the small sample size and the relatively short follow-up period, due to the recent adoption of ESL and the rarity of CSL. However, the recent period analyzed ensures a homogeneous population, treated by the same surgical group, with the same indications, surgical techniques and treatment patterns, without historical bias, and provides insight into what can actually be achieved in a high-volume unit. The lack of a standardized assessment of quality of life (QoL), postoperative spirometry [29] and functional assessment over time could be considered another limitation.

## 5. Conclusions

Complex sleeve lobectomies are an effective option in the treatment of centrally located NSCLC and ensure a high rate of complete resection with fewer major postoperative complications and lower mortality compared to PN. Neoadjuvant treatment is not associated with an increased risk of serious and bronchial adverse events, so CSL should also be considered safe as part of a multimodality treatment pathway. Although the local recurrence rate after CSL is higher compared to PN, this had no impact on overall survival and disease-free survival, so CSL should be considered the procedure of choice not only in patients with limited pulmonary reserve, ruling out PN, but also in all patients in whom complete resection can be guaranteed.

## Figures and Tables

**Figure 1 cancers-16-00261-f001:**
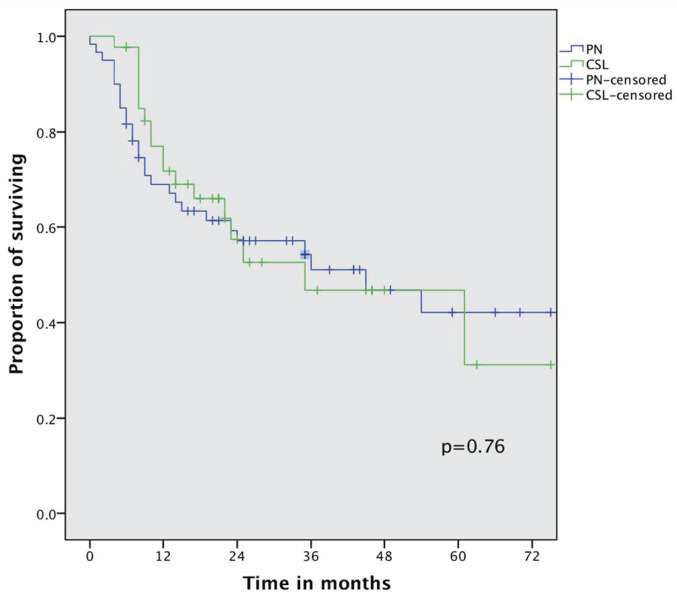
Overall survival curve of the whole cohort comparing pneumonectomy (PN) and complex sleeve lobectomy (CLS).

**Figure 2 cancers-16-00261-f002:**
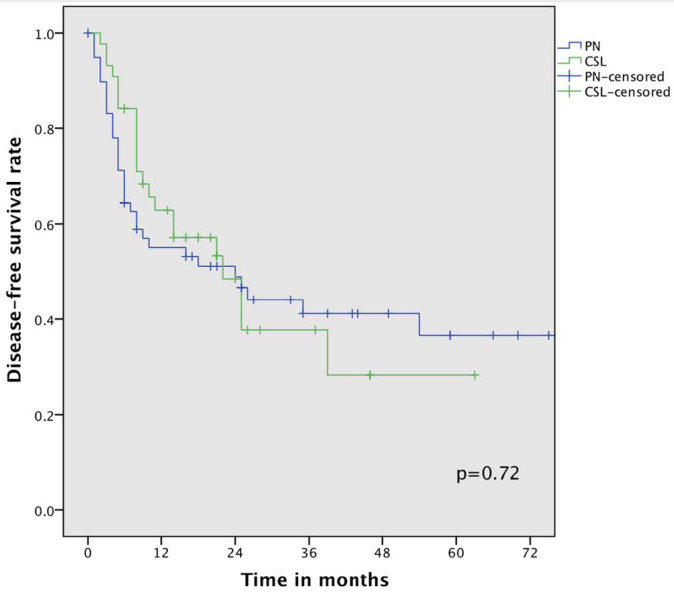
Disease-free survival curve comparing pneumonectomy (PN) and complex sleeve lobectomy (CSL).

**Figure 3 cancers-16-00261-f003:**
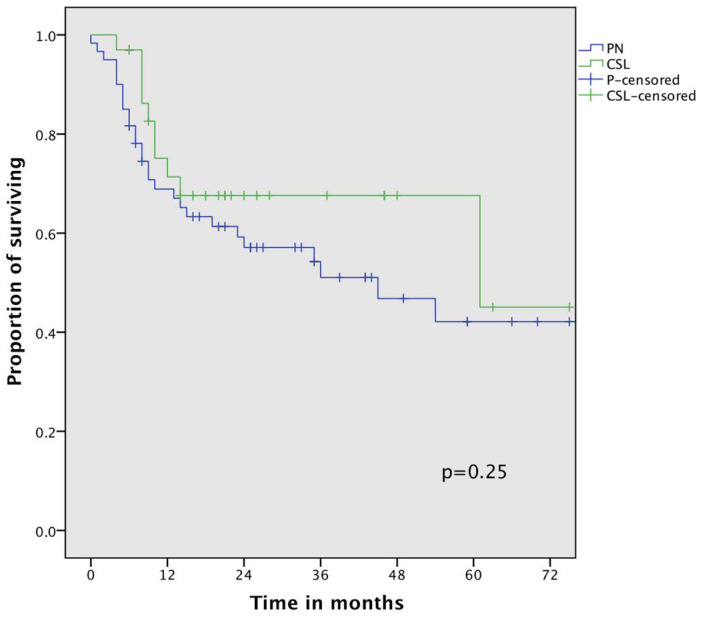
Overall survival curve of non-compromised patients comparing pneumonectomy (PN) and complex sleeve lobectomy (CSL).

**Table 1 cancers-16-00261-t001:** Demographic, preoperative data and clinical stages (PN: pneumonectomy; CSL: complex sleeve lobectomy; PS ECOG: Performance Status Eastern Cooperative Oncological Group; ASA: American Society of Anesthesiologists; mCCI: modified Charlson Comorbidity Index; FEV1: forced expiratory volume in 1 s; DLCO: diffusing capacity of the lung for carbon monoxide; NAC: neoadjuvant chemotherapy; NACRT: neoadjuvant chemoradiotherapy).

Variable	PN (n = 60)	CSL (n = 44)	*p*
Sex male	40 (66.7%)	31 (70.5%)	0.68
Age	66.6 ± 8.01	67.6 ± 6.73	0.53
BMI	25.8 ± 4.6	24.1 ± 3.8	0.06
PS			0.8
0	16 (26.7%)	13 (29.5%)	
1	33 (55%)	25 (56.8%)	
2	11 (18.3%)	6 (13.6%)	
mCCI			0.65
0	13 (21.7%)	9 (20.5%)	
1	12 (20%)	11 (25%)	
2	15 (25%)	15 (34.1%)	
3	10 (16.7%)	3 (6.8%)	
>4	10 (16.7%)	6 (13.7%)	
ASA			0.42
1	13 (21.7%)	11 (25%)	
2	22 (36.7%)	19 (43.2%)	
3	25 (41.7%)	13 (29.5%)	
4	0	1 (2.2%)	
FEV1%	81.7 ± 16	82.2 ± 20.4	0.9
FVC%	95.3 ± 18.7	99.9 ± 24	0.33
DLCO%	68.5 ± 17.9	66.7 ± 15.5	0.63
cSTAGE			0.64
IA	3 (5%)	1 (2.3%)	
IB	1 (1.7%)	0	
IIA	2 (3.3%)	2 (4.5%)	
IIB	15 (25%)	13 (29.5%)	
IIIA	33 (55%)	22 (50%)	
IIIB	4 (6.7%)	6 (13.6%)	
IIIC	1 (1.7%)	0	
IV	1 (1.7%)	0	
NAC	21 (35%)	14 (31.8%)	0.73
NACRT	0	2 (4.9%)	0.34

**Table 2 cancers-16-00261-t002:** Surgical procedures performed. Complex sleeve lobectomies are classified into extended sleeve lobectomy (according to the modified Okada classification [12,14]) and double-sleeve lobectomy (PN: pneumonectomy; CSL: complex sleeve lobectomy; ESL: extended sleeve lobectomy).

Variable	PN (n = 60)	CSL (n = 44)	*p*
Type of procedure	Right 23 (38.3%) Left 37 (61.7%)	ESL type A 15 (34.1%)	
		ESL type B 2 (4.5%)	
		ESL type C 17 (38.6%)	
		ESL type D 3 (6.8%)	
		ESL not classified (E) 1 (2.3%)	
		Double sleeve 6 (13.5%)	
Bronchial flap coverage	33 (55%)	5 (11.4%)	<0.01

**Table 3 cancers-16-00261-t003:** Postoperative, pathological data, pattern of recurrence and type of postsurgical treatment (ADC: adenocarcinoma; SCC: squamous cell carcinoma; HS: hospital stay; ACHT: adjuvant chemotherapy; ART: adjuvant radiotherapy).

Variables	PN (n = 60)	CSL (n = 44)	*p*
Pathology			0.14
ADC	29 (48.3%)	15 (34.1%)	
SCC	31 (51.7%)	29 (65.9%)	
pSTAGE			0.71
0	0	1 (2.3%)	
IA	3 (5%)	3 (6.8%)	
IB	4 (6.7%)	1 (2.3%)	
IIA	4 (6.7%)	3 (6.8%)	
IIB	11 (18.3%)	11 (25%)	
IIIA	21 (35%)	16 (36.4%)	
IIIB	17 (28.3%)	9 (20.5%)	
pN0	20 (33.3%)	13 (29.5%)	0.78
pN1	18 (30%)	16 (36.4%)	
pN2	22 (36.7%)	15 (34.1%)	
pR0	53 (88.3%)	39 (88.6%)	0.96
Tumor diameter cm	5.74 ± 2.76	5.14 ± 2.85	0.3
HS	10.68 ± 7.1	12.7 ± 8.93	0.18
Patients with at least one complication	36 (60%)	24 (54.5%)	0.57
Clavien–Dindo classification			<0.01
1	2 (3.3%)	1 (2.3%)	
2	21 (35%)	8 (18.2%)	
3a	0	11 (25%)	
3b	3 (5%)	1 (2.3%)	
4a	6 (10%)	2 (4.5%)	
4b	1 (1.7%)	0	
5	3 (5%)	0	
Major complications (>3b)	13 (21.7%)	3 (6.8%)	0.038
Bronchial dehiscence	4 (6.7%)	2 (4.5%)	0.64
Postoperative mortality	3 (5%)	0	0.13
Recurrence	26 (43.3%)	20 (45.5%)	0.83
Pattern of recurrence			0.1
No recurrence	34 (56.7%)	24 (54.5%)	
Local	1 (1.7%)	6 (13.6%)	
Regional	3 (5%)	2 (4.5%)	
Distant	22 (36.7%)	12 (27.3%)	
ACHT	27 (45.8%)	19 (43.2%)	0.79
ART	6 (10.9%)	2 (7.1%)	0.58

**Table 4 cancers-16-00261-t004:** Univariate and multivariable logistic regression analysis of major complications (OR: odds ratio; CI95%: confidence interval 95%; PN pneumonectomy; CSL: complex sleeve lobectomy; ECOG PS: Eastern Cooperative Oncology Group Performance Status; mCCI: modified Charlson Comorbidity Index; FEV1%: forced expiratory volume at the first second; DLCO%: diffusion capacity of the lung for carbon monoxide; NAC: neoadjuvant chemotherapy).

Univariate Analysis				Multivariable Analysis		
Variable	OR	CI95%	*p*	OR	CI95%	*p*
Sex male	1.85	0.62–5.5	0.26			
Age > 70	2.16	0.68–6.87	0.18	3.31	0.86–12.7	0.081
PN	3.78	1.006–14.2	0.049	4.21	0.93–19.03	0.073
ECOG PS > 2	2.87	0.85–9.75	0.089	2.07	0.32–13.2	0.44
mCCI > 3	2.33	0.77–7	0.13	1.59	0.29–8.51	0.58
FEV1% < 60	1.42	0.16–12.5	0.75			
DLCO% < 60	1.76	0.5–6.19	0.37			
NAC	1.63	0.48-5.48	0.42			
Clinical stage						0.14
I–II	ref					
III–IV	2.25	0.76–6.61	0.14	2.75	0.7–10.7	
Right side	3.85	1.22–12.07	0.021	5.75	1.43–23.1	0.014

**Table 5 cancers-16-00261-t005:** Univariate and multivariable Cox regression analysis of overall survival (CSL: complex sleeve lobectomy; PN pneumonectomy; ECOG PS: Eastern Cooperative Oncology Group Performance Status; ACHT: adjuvant chemotherapy; ART: adjuvant radiotherapy).

Univariate Analysis				Multivariable Analysis		
Variable	HR	CI95%	*p*	HR	CI95%	*p*
Sex male	1.21	0.66–2.19	0.52			
Age > 70	1.65	0.88–3.06	0.11	1.71	0.72–4.06	0.22
PN	0.97	0.53–1.76	0.92			
CLS	ref					
ECOG PS > 2	2.23	0.87–5.6	0.09	1.78	0.6–5.25	0.29
R1	1.41	0.78–2.53	0.25			
Stage III–IV	1.49	0.8–2.77	0.2			
Recurrence	5.47	2.7–11	<0.01	6.88	2.5–18.9	<0.01
Pathology						
SCC	Ref					
ADC	1.06	0.59–1.91	0.82			
ACHT	0.82	0.45–1.48	0.52			
ART	3.19	1.27–8.01	0.014	2.53	0.88–7.25	0.084

**Table 6 cancers-16-00261-t006:** Pros and cons comparison table between PN and CSL.

**Pneumonectomy**	
**PROS**	**CONS**
Technically easier	Higher mortality
Lower local recurrence	Higher major complication rate
	Complex management of complications
	No survival advantage
	Theoretically worse QoL, worse compliance with other oncological treatments
**Complex sleeve lobectomy**	
Parenchymal preservation, theoretically more functional conservation	More technically demanding procedure
Lower incidence of major complications	Higher local recurrence rate
Low mortality	
Resection also feasible in compromised patients	
Easier management of complications	

## Data Availability

The data presented in this study are available on request from the corresponding author. The data are not publicly available due to privacy.

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
