# Peer review of "Complex Sleeve Lobectomy Has Lower Postoperative Major Complications Than Pneumonectomy in Patients with Centrally Located Non-Small-Cell Lung Cancer"

_cancers, 2024, doi:10.3390/cancers16020261_

Round 1
Reviewer 1 Report
Comments and Suggestions for Authors
Work well done, only a few inaccuracies need to be corrected and the bibliography expanded a little.
The work is interesting but there are some considerations to make.
1) Make sure that the aim of the introduction corresponds to the primary end-secondary points of the materials and methods.
2) Table 2: expand the caption title.
3) Line 185: when talking about mortality in the PN group which is 5% vs 0% it must be emphasized that the value is not statistically significant.
4) Line 197: the univariate analysis must be described first and then the multivariate one and only the statistically significant values must be emphasized.
5) Line 215: clarify better that overall survival at 3 years is better for the CSL group (although not statistically significant) and that for the CSL group there are more local recurrences, while for the PN group distant recurrences.
6) Line 237: We must also comment on the multivariate.
7) Line 257: I would change the sentence to say that CSL appears (in terms of probability) to be a sound and effective surgical procedure…
8 References: if possible expand the bibliographic references in the discussion because only 3 entries seem few.
Reviewer 2 Report
Comments and Suggestions for Authors
1- The manuscript needs comprehensive writing revision
2- Some abbreviations are reported twice, please revise
i.e "complex sleeve lobectomies (CSL)"
3- It would be good to highlight the main objectives of this study and its findings clealry.
4- Table 1, 3, 4: move the appreciation down the table with a smaller font
5- In line 258 you say "We demonstrated that CSL is a safe and effective surgical procedure for 257 centrally located NSCLC with superior post-operative results and equivalent long-term 258 outcomes compared to pneumonectomy". In line 266 and "we observed a lower 90-day mortality for CSL, although 266 not statistically significant compared to PN" (in line 273-274 as well) can you explain this?
6- 282 add a reference
7- You can draw a comparison table demonstrating the cons and pros of these the techniques based on the investigation you did.
Comments on the Quality of English LanguageModerate editing of English language required
